# Preparations of Dutch emergency departments for the COVID-19 pandemic: A questionnaire-based study

**Rory D. O'Connor**[1]*, **Dennis G. Barten**[2], **Gideon H. P. Latten**[3]

**1** Department of Emergency Medicine, Jeroen Bosch Hospital, 's-Hertogenbosch, The Netherlands, **2** Department of Emergency Medicine, VieCuri Medical Center, Venlo, The Netherlands, **3** Department of Emergency Medicine, Zuyderland Medical Center, Heerlen, The Netherlands

* r.o.connor@jbz.nl

**Data Availability Statement:** All relevant data are within the manuscript and its Supporting Information files.

## Abstract

### Background

The onset of the COVID-19 pandemic was characterized by rapid increases in Emergency department (ED) patient visits. EDs required an appropriate transformation. The main challenges were: adapting capacity to respond to surges in the number of patient visits, protection of high risk (frontline) staff and the segregation of suspect-COVID-19 patients. To date, only a few studies have assessed the nation-wide response of EDs to the COVID-19 pandemic. This study was designed to review the preparations of Dutch EDs during the initial phase of this public health crisis.

### Methods

The study was designed as a nation-wide, cross-sectional, questionnaire-based study of Dutch hospital organizations having one or more EDs. One respondent completed the questionnaire for each hospital. The questionnaire was conducted between the first and the second COVID-19 wave in the Netherlands. It contained close-ended and open-ended questions on changes in ED infrastructure, ED workforce adaptions and the role of emergency physicians (EPs) in each hospital crisis management team.

### Results

The questionnaire was completed by 58 respondents. This represented 80% of the total number of EDs. All respondents had made preparations in anticipation of a COVID-19 patient surge. Treatment capacity was expanded in 70% of EDs, with a median increase of 49% (IQR 33–73%). Suspect-COVID-19 was segregated from non-COVID-19 patients in 89% of EDs. Alternative locations (such as outpatient departments) were more often used to assess non-COVID-19 patients, than for suspect-COVID-19 patients. Staff was expanded in 82% of EDs. This largely concerned nursing staff. A formal role for Emergency Physicians (EPs) in the hospital's crisis management team was reported by 94% of hospital organizations employing EPs.

**Funding:** The author(s) received no specific funding for this work.

**Competing interests:** The authors have declared that no competing interests exist.

**Abbreviations:** EP, Emergency physician; ED, Emergency department; ICU, Intensive care unit; IQR, Interquartile Range.

## Conclusion

All Dutch EDs responded to the COVID-19 pandemic in a very short time span despite much uncertainty. Preparations predominantly concerned expansion of treatment capacity and segregation of COVID-19 ED care. EPs played a prominent role, both in direct COVID-19 care and in the hospital crises management team. It is vital for EDs to adapt to community needs swiftly. The ability of EDs to respond to the pandemic varied considerably.

## Introduction

Coronavirus disease 2019 (COVID-19), which emerged in Southeast China in December 2019, was declared a pandemic by the World Health Organization on March 11, 2020 [1]. As it spread rapidly around the globe, hospital emergency departments (EDs) braced for impact.

In the Netherlands, the first case of COVID-19 was identified on February 27, 2020 [2]. As of May 31, 2020, which can be considered the end of the first Dutch COVID-19 wave, there were 45,976 confirmed cases of infection (of which 11,674 were hospitalized) and 5,939 confirmed COVID-19 deaths [3]. It resulted in a national incidence of 264 cases per 100,000 inhabitants. The most affected region, situated in the south, registered an incidence of 501 compared to an incidence of 60 per 100,000 inhabitants for the least affected region in the north of the Netherlands.

Emergency medical services and hospital EDs are viewed as public health services that are responsible for the initial medical response to any type of disaster, both in the short and long term [4]. In contrast to sudden-onset events, large-scale infectious outbreaks typically require a prolonged, sustained response [4, 5]. The current COVID-19 pandemic was initially characterized by rapidly increasing patient hospital contacts. A swift overhaul of several aspects of ED preparations in Dutch hospitals was imperative [6, 7]. Challenges mainly concerned surge capacity, frontline staff (staff at high risk of infection during initial contact with patients) protection and the segregation of suspect-COVID-19 patients [8–11].

To date, few studies have assessed nation-wide ED ability to cope with the COVID-19 pandemic. A French questionnaire-based study, conducted during an early stage of the pandemic (March 7 to March 11, 2020), revealed that EDs were poorly prepared [12]. A similar study from India, limited to academic EDs, showed that 90% of hospitals had developed specific COVID-19 triage systems and that almost 80% established dedicated areas for suspect-COVID-19 patients. However, it also revealed that the level of preparation of EDs varied widely. The authors stated that an individualized coping strategy for each ED which considers baseline needs and available resources is superior to a blanket strategy applied to all EDs. Although this claim seems sensible, evidence is scant [13].

Whilst clinical and intensive care unit (ICU) capacity for COVID-19 in Dutch hospitals were closely monitored and controlled through a national body (Landelijk Coördinatiecentrum Patiënten Spreiding), there was no guidance on the surge capacity management of EDs [6]. Consequently, hospitals largely restructured the organization of their EDs on an individual basis. This study aimed to form an overview of preparations that were taken in Dutch EDs during the initial phase of the COVID-19 pandemic. In addition, it aimed to explore the role of Dutch emergency physicians (EPs) in the hospitals' crisis management teams.

## Methods

### Setting

The Dutch healthcare system is modern. It has an effective primary care system and a finely meshed network of specialized acute and critical care facilities. This network includes 82 EDs (Fig 1), which are located within 71 hospital organizations. 10 of these hospital organizations have multiple ED locations. The EDs serve the Dutch population of 17.4 million people. On average an individual ED is attended by 22,500 patients per year, of whom 17% are self-referrals [14].

### Study design

This study was designed as a nation-wide, cross-sectional, questionnaire-based study of Dutch hospital organizations with one or more EDs. For each hospital one respondent, either an EP or an ED manager, received an invitation to participate by email on July 29, 2020. The Netherlands Society of Emergency Physicians distributed the invitations. It has a database of all Dutch EDs. Contact details remained anonymous to the researchers in compliance with laws for the protection of personal information.

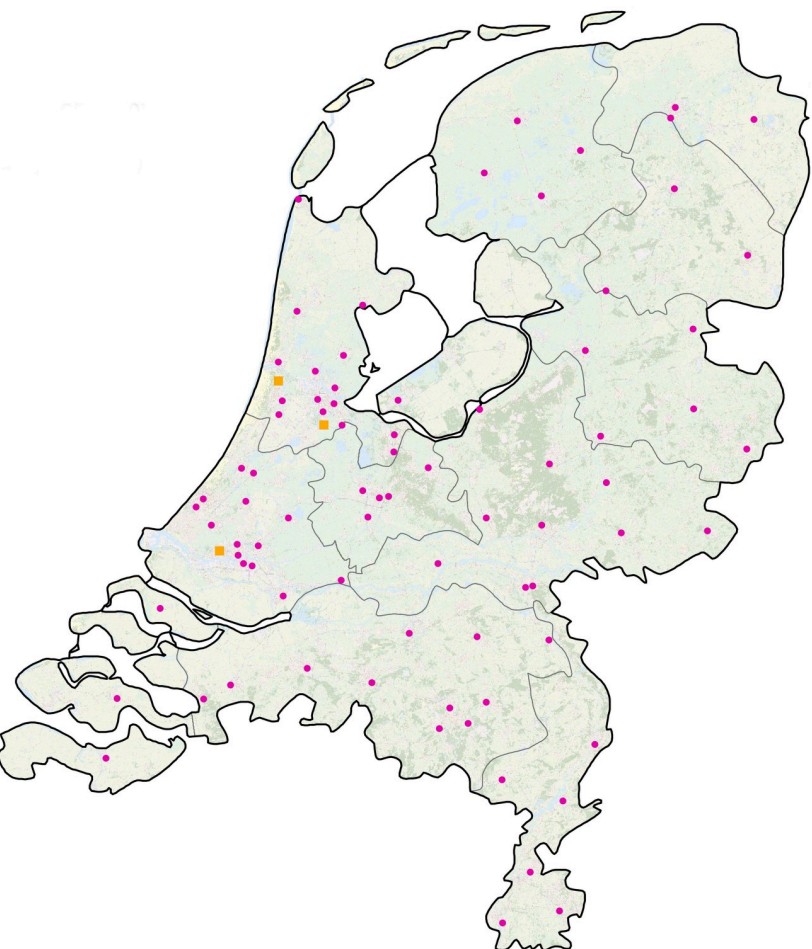

**Fig 1. Emergency departments in the Netherlands (June 2020) [15].** Pink circle: Opened 24 hours, 7 days a week. Yellow square: Opened day and evening, 7 days a week.

If a respondent did not complete the questionnaire, a reminder was sent every fortnight. The questionnaire could be completed until September 30, 2020. Respondents were requested to complete the questionnaire on behalf of the hospital in which they practiced. When a hospital had multiple EDs, the questionnaire facilitated provision of information on all EDs.

The questionnaire contained 15 close-ended questions (dichotomous and multiple-choice) and 2 open-ended questions. Broadly, all these questions covered 3 topics: changes in ED infrastructure, alterations in ED workforce and the role of EPs in the hospital's crisis organization. A Dutch and an English version of the questionnaire is supplied as S1 and S2 Files. Crowding was defined as described by Asplin et al. [16].

## Statistical analysis

All analyses were performed with SPSS version 26 (SPSS Inc., Chicago, USA). Continuous data were reported as means with standard deviation (SD) or as medians with interquartile ranges (IQR). Categorical data were reported as absolute numbers and as valid percentages (to correct for missing data).

All data was collected anonymously. The Strengthening the Reporting of Observational Studies in Epidemiology guidelines was used for reporting this observational study [17]. The Medical Ethics Committee Zuyderland & Zuyd concluded that the rules of the Medical Research Involving Human Subjects Act (WMO in Dutch) do not apply to this study (METCZ20200130). The study was registered in the Netherlands Trial Register (Trial number NL8818).

## Results

The questionnaire was completed on behalf of 66 (80%) out of 82 EDs (Table 1). These EDs served 58 (82%) out of 71 hospital organizations, as eight hospital organizations had multiple ED locations. Prior to the COVID-19 pandemic, the majority of the EDs had an annual attendance of less than 30.000 patients per year. 86% of EDs were staffed by EPs.

**Table 1. Baseline ED characteristics.**

|  | EDs (n = 66)* | EDs that did not respond (n = 16)* |
|---|---|---|
| Annual attendance |  |  |
| <20,000 patients | 19 (28%) |  |
| 20,000–25,000 patients | 17 (25%) |  |
| 25,000–30,000 patients | 13 (19%) |  |
| 30,000–35,000 patients | 8 (12%) |  |
| 35,000–40,000 patients | 3 (4%) |  |
| >40,000 patients | 6 (9%) |  |
| Staffed by EPs | 57 (86%) | 10 (63%) |
| Preparations made for COVID-19 pandemic | 66 (100%) |  |
| Incidence** in region where ED is situated |  |  |
| < 100 | 6 (9%) | 1 (6%) |
| 100–200 | 13 (20%) | 2 (13%) |
| 200–300 | 20 (30%) | 8 (50%) |
| 300–400 | 21 (32%) | 5 (31%) |
| > 400 | 6 (9%) | 0 |

Abbreviations: ED–emergency department, EPs–emergency physicians.

* Data are presented as n (%).

** COVID-19 cases per 100.000 as of May 31, 2020 [3].

The majority of EDs were situated in regions with 100 to 400 COVID-19 cases per 100,000 inhabitants This was the case for EDs that completed the questionnaire (82%) and EDs that did not respond (94%). The 16 EDs that did not respond were localized in different regions throughout the Netherlands and included university, teaching and peripheral hospitals [3].

All participating EDs had made preparations in anticipation to a surge of COVID-19 patients. The date when these preparations were completed varied between February 24 and April 1, 2020 (Median: 16 March 2020; IQR: 11–21 March 2020).

## Changes in ED infrastructure

Before the COVID-19 pandemic, the median number of ED treatment spaces was 17 (IQR 12–21) (Table 2). Treatment capacity was expanded in 46 (70%) EDs. The median number of additional treatment spaces was 8 (IQR 4–10). This is a median increase of 49% (IQR 33–73%).

Reasons for not increasing the ED area included: a previous reduction of ED utilization for several logistic alterations (15%), the ED being designated as a non-COVID-19 ED (6%), and the inability to expand ED treatment spaces due to isolation measures demanding more space per patient (4.5%).

Logistic alterations to standard practice included: the redirection of less urgent ED visits, such as minor traumatic injuries, to outpatient departments in 42 (63%) EDs and 12 (18%)

**Table 2. Changes in ED infrastructure.**

|  | EDs (n = 66)* | Number of Spaces (IQR)* |
|---|---|---|
| Number of treatment spaces was increased during pandemic | 46 (70%) | |
| - Pre-pandemic treatment spaces | | 17 (12–21) |
| - Additional treatment spaces | | 8 (4–10) |
| Number of treatment spaces was not increased during pandemic | 20 (30%) | |
| Reasons for not increasing treatment spaces: | | |
| ○Logistical alterations to usual ED practice | 10 (15%) | |
| ○non-COVID-19 ED | 4 (6%) | |
| ○Expansion not feasible | 3 (4%) | |
| ○Other | 3 (4%) | |
| COVID-19 ED care was segregated from non-COVID-19 ED care | 59 (90) | |
| Location of COVID-19 ED care | | |
| - Original ED only | 39 (59%) | |
| - Original ED and other location | 14 (21%) | |
| - Other location only | 7 (11%) | |
| Location of non-COVID-19 ED care | | |
| - Original ED only | 25 (38%) | |
| - Original ED and other location | 27 (41%) | |
| - Other location only | 7 (11%) | |
| Screening for COVID-19 before ED entry performed with | | |
| - Symptom-based screening list only | 43 (65%) | |
| - Symptom-based screening list and radiological imaging (Chest X-ray or CT) | 13 (20%) | |
| - Chest CT only | 1 (2%) | |

ED–emergency department.

* Data are presented as median (IQR), or n (%).

EDs actually effectuated a faster admission process to hospital wards and intensive care units. The latter resulted in a shortened length of stay in the ED.

Suspect-COVID-19 patients were segregated from non-COVID-19 patients in 59 (89%) EDs. In the majority (59%) of EDs, this was organized within the original ED allotted area. Alternative locations used by the remaining EDs can be found in S1 Table. In most (75%) EDs, a symptom-based checklist alone was used to assign a suspicion of COVID-19 infection.

In 46 (79%) hospital organizations, one or more of the measures implemented for the pandemic were intended as permanent (S2 Table). These included improved infection prevention in 13 (22%), improved interdisciplinary collaboration in 13 (22%), permanent adjustments to segregate infectious patients in 10 (17%) and permanent redirection of less urgent patients in 8 (14%) hospital organizations.

## Alterations in ED workforce

In 54 (82%) EDs the workforce was modified (Table 3). Nursing staff was expanded by redeploying, both additional specialized ED nurses (53%), and nursing staff from other departments (61%) took part. A large variety of physicians took part directly in COVID-19 ED care. Emergency medicine (86%), internal medicine (85%) and pulmonology (82%) were involved most frequent. In 21 (32%) EDs, the additional workforce consisted of nurses and physicians only. In the remaining 45 (68%) EDs other medical disciplines were also deployed.

## Role of EPs in the crisis organization

At 49 (85%) hospital organizations EPs were employed. In these hospital organizations EPs were directly involved in the assessment and treatment of COVID-19 patients. They had an additional coordinating role in the ED in 43 (88%) and they were involved in triage or segregation of suspect-COVID-19 patients in 40 (82%) hospital organizations. A formal role of EPs in the hospital's crisis management team was reported in 46 (94%) hospital organizations. An EP was member of the strategic crisis team in 19 (39%) and of the operational crisis management team in 32 (65%) hospital organizations.

## Crowding

The majority (52%) of hospital organizations experienced no crowding during the first COVID-19 surge [16]. Occasional crowding was reported by 24 (41%). Four (7%) hospital organizations experienced crowding multiple times a week.

## Discussion

This questionnaire-based study aimed to provide an overview of preparations of Dutch EDs for the initial phase of the COVID-19 pandemic. With a high response rate of 80% of EDs, the results are representative for the majority of Dutch EDs.

All participating EDs made preparations for a surge in COVID-19 patients. Treatment capacity area was increased in almost 70% of the participating EDs, with a median increase in treatment spaces of 50%. Suspect-COVID-19 patients were segregated from non-COVID-19 patients in 89% of EDs. The ED workforce was expanded in 82% of EDs. EPs were directly involved in the care for COVID-19 patients in all EDs and they had a prominent role in the hospital crisis management team in 94%.

The COVID-19 pandemic obliged EDs to make drastic organizational changes in a very short time span. It was then unclear for EDs if they would be adequately compliant for the requirements of the pandemic or if they were even necessary [8]. There is national Dutch

**Table 3. Alterations in ED workforce.**

| | EDs* |
|---|---|
| Expansion of nursing staff | 54 (82%) |
| Additional ED nurses | 35 (53%) |
| Additional non-ED nurses | 40 (61%) |
| Specialties involved in ED COVID-19 care | |
| - Emergency medicine | 57 (86%) |
| - Internal medicine | 56 (85%) |
| - Pulmonology | 54 (82%) |
| - Anesthesiology | 26 (40%) |
| - Geriatrics | 24 (36%) |
| - Surgery | 23 (35%) |
| - Neurology | 22 (33%) |
| - Cardiology | 20 (30%) |
| - Pediatrics | 20 (30%) |
| - Gastroenterology | 16 (24%) |
| - Orthopedics | 14 (21%) |
| - Otolaryngology/ENT | 12 (18%) |
| - Urology | 12 (18%) |
| - Dermatology | 6 (9%) |
| - Primary care | 6 (9%) |
| - Plastic surgery | 6 (9%) |
| - Rheumatology | 5 (8%) |
| - Gynecology | 4 (6%) |
| - Other | 18 (27%) |
| Other disciplines | 44 (67%) |
| - Medical interns | 17 (26%) |
| - Physician assistants | 15 (23%) |
| - Doctor's assistants | 15 (23%) |
| - Surgery assistants | 14 (21%) |
| - Anesthetic nurses | 9 (14%) |
| - Orthopedic practitioner | 9 (14%) |
| - Other** | 7 (11%) |

ED–emergency department, ENT–ear nose throat.

* Data are presented as n (%).

** Volunteers, medical students.

guidance on clinical and ICU capacity [6]. Remarkably, however there was then no consensus or general advice on ED capacity. This is reflected by the differences found between responding EDs in this study.

Some standardization of EDs may be indeed desirable. Nonetheless most EDs planned their surge response both individually and to their satisfaction. The majority reported only occasional or no crowding. In this perspective, it is important to acknowledge that ED surge capacity planning should take individual hospital characteristics into consideration. Indeed, improvisation can be important, even when there are national guidelines. During the first wave of the COVID-19 pandemic, regions within the Netherlands differed considerably with regards to COVID-19 infection rates. This may have influenced the workload of some EDs. It could in part explain the differences between ED pandemic approaches.

The COVID-19 pandemic may have changed ED care forever and some adaptions in EDs have become permanent. E-health applications have flourished and there is more focus on securing optimal care at the correct institution [18]. Not all patient categories need ED care, but may continue to receive safe and efficient care at another location.

Furthermore, this public health crisis has shown the importance of a strong emergency and critical care system. A certain degree of overcapacity may be pivotal for an effective response. As this pandemic is ongoing, surge capacity models that allow some flexibility may be the most useful [7, 9, 19]. Hospital capacity is dynamic and highly dependent on the occupancy of available resources [20]. When the pressure on ED care is lower, capacity could be used for non-urgent care and vice versa. This way, EDs could comply timely with community demands.

Close collaboration within EDs has always been of vital importance. As shown by our results, virtually all medical disciplines were deployed in the EDs during the pandemic. Although this survey did not examine the quality of inter-disciplinary collaboration, multiple respondents greatly valued the unique situation where all kinds of disciplines worked closely together. It may not come as a surprise that EPs, internists, and pulmonologists were involved in COVID-19 ED care. However, EPs also played an important role in ED coordination and triage. Furthermore, EPs played a vital role in the hospitals' crisis management teams. This emphasizes the necessity of the inclusion of experienced staff members working specifically in the ED.

As outbreaks of novel infectious diseases share similar characteristics, the results of this study may also relate to other future pandemics. Usually, little is known about the pathophysiology, symptomatology, and contagiousness of the disease. The lack of knowledge compels EDs to have a low threshold for isolating patients who might be contagious. EDs should therefore invest in isolation capacity. Furthermore, pandemics may result in high numbers of patients who require emergency care, which underlines the need for health care systems to have sufficient surge capacity. Finally, alterations of usual care, such as the redirection of low-acuity ED patients to outpatient departments, may help to alleviate the pressure on EDs during future pandemics.

This study does have limitations. Firstly, this was a retrospective questionnaire-based study filled in by one respondent per ED. Researchers were unaware who the anonymous respondents were and did not know the extent of their involvement in the hospital's crisis management team. Also, the extent to which respondents were aware of the issues of care at their own hospital was not clear. Furthermore, a non-responder bias could exist, though the response rate was high and the regional COVID-19 rates of the EDs without a response were similar to the participating EDs. Lastly, patients' self-report could have affected the validity of responses.

Globally emergency services were used less frequently during the pandemic [21]. This phenomenon is not yet completely understood. However, it may have protected many EDs from overcrowding despite their maintenance of full non-COVID-19 ED care. The pandemic approaches of these EDs may not be as successful in other crisis situations. Finally, the results of this study may not apply to EDs in some healthcare systems. This is the case in those without a strong primary care system functioning as gatekeepers for the EDs. In the Netherlands, a relatively large proportion of ED patients (82%) is referred by a general practitioner or by emergency medical services [14].

## Conclusion

This study showed that all Dutch EDs made preparations for COVID-19 in a short time span and despite many uncertainties. Preparations primarily included the expansion of treatment

capacity and the segregation of COVID-19 care. EPs had a prominent role, both in direct patient COVID-19 ED care and in the crisis management teams of hospitals. It is vital for EDs to be able to adapt in response to community requirements. The ability of ED's to achieve this during the pandemic varied considerably.

## Supporting information

**S1 File. Questionnaire English.**
(DOCX)

**S2 File. Questionnaire Dutch.**
(DOCX)

**S1 Table. Alternative locations of emergency care.**
(DOCX)

**S2 Table Measures intended as permanent.**
(DOCX)

**S1 Dataset. Emergency departments.**
(XLSX)

**S2 Dataset. Hospital organisations.**
(XLSX)

## Acknowledgments

We would like to thank all participating EDs for their participation in this study. We also want to thank the Dutch Society of Emergency Physicians for its support in distributing the questionnaire.

## Collaborators

L.M. Esteve Cuevas, M.L. Ridderikhof, dr W.A.M.H. Thijssen, R.R. Pigge, N.E. Mullaart-Jansen, R.J.C.G. Verdonschot, V. Brown, G. van Woerden, E.L. Janssens, B.Y.M. van der Kolk, B. de Groot, F. Derkx-Verhagen, W.P. Poortvliet, H. Lameijer, Y. Schoon, J. Holkenborg, L.E. Kerkvliet, M.S.A. de la Fosse, E. ter Avest, MD, PhD., K. Azijli, S. Postma, J.M. van Lieshout, B. Vlaming, C. Kok, M. Maltha, R. Lulf, R.J.L. Boden, A.E. Boendermaker, J.L.P. Kuijten, J.L. van der Meer, K. van den Broek, L. Jansen, M.J. Meijer, T.B. Nanlohij, D.J.R. Keereweer, A.G. Pol, T.J. Oosterveld-Bonsma, J.M. Huttenhuis, G.B. Spijkers, J. Jaspers.

## Author Contributions

**Conceptualization:** Gideon H. P. Latten.

**Data curation:** Rory D. O'Connor, Dennis G. Barten, Gideon H. P. Latten.

**Formal analysis:** Rory D. O'Connor, Dennis G. Barten, Gideon H. P. Latten.

**Investigation:** Rory D. O'Connor, Dennis G. Barten.

**Methodology:** Rory D. O'Connor, Dennis G. Barten, Gideon H. P. Latten.

**Project administration:** Rory D. O'Connor.

**Validation:** Dennis G. Barten.

**Visualization:** Gideon H. P. Latten.

**Writing – original draft:** Rory D. O'Connor, Dennis G. Barten, Gideon H. P. Latten.

**Writing – review & editing:** Rory D. O'Connor, Dennis G. Barten, Gideon H. P. Latten.

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
