## [Decision Letter · Decision Letter 0]

2 Jul 2021

PONE-D-21-12869

Preparations of Dutch emergency departments for the COVID-19 pandemic: a questionnaire-based study.

PLOS ONE

Dear Dr. O'Connor,

Thank you for submitting your manuscript to PLOS ONE. After careful consideration, we feel that it has merit but does not fully meet PLOS ONE’s publication criteria as it currently stands. Therefore, we invite you to submit a revised version of the manuscript that addresses the points raised during the review process.

Please address the issues and revise accordingly.

We look forward to receiving your revised manuscript.

Kind regards,

Academic Editor

PLOS ONE

Journal Requirements:

2. We note that Figure 1 in your submission contain map images which may be copyrighted. All PLOS content is published under the Creative Commons Attribution License (CC BY 4.0), which means that the manuscript, images, and Supporting Information files will be freely available online, and any third party is permitted to access, download, copy, distribute, and use these materials in any way, even commercially, with proper attribution. For these reasons, we cannot publish previously copyrighted maps or satellite images created using proprietary data, such as Google software (Google Maps, Street View, and Earth). For more information, see our copyright guidelines: http://journals.plos.org/plosone/s/licenses-and-copyright.

2.1.    You may seek permission from the original copyright holder of Figure 1 to publish the content specifically under the CC BY 4.0 license. 

2.2.    If you are unable to obtain permission from the original copyright holder to publish these figures under the CC BY 4.0 license or if the copyright holder’s requirements are incompatible with the CC BY 4.0 license, please either i) remove the figure or ii) supply a replacement figure that complies with the CC BY 4.0 license. Please check copyright information on all replacement figures and update the figure caption with source information. If applicable, please specify in the figure caption text when a figure is similar but not identical to the original image and is therefore for illustrative purposes only.

Reviewers' comments:

Reviewer's Responses to Questions

**Comments to the Author**

1. Is the manuscript technically sound, and do the data support the conclusions?

Reviewer #1: Partly

Reviewer #2: Yes

2. Has the statistical analysis been performed appropriately and rigorously? 

Reviewer #1: Yes

Reviewer #2: Yes

3. Have the authors made all data underlying the findings in their manuscript fully available?

Reviewer #1: No

Reviewer #2: Yes

4. Is the manuscript presented in an intelligible fashion and written in standard English?

Reviewer #1: Yes

Reviewer #2: Yes

5. Review Comments to the Author

Reviewer #1: In their manuscript, O'Connor et al describe the results of a questionnaire filled out by individuals involved in the management of Dutch emergency departments, specifically on the preparations involved during the first wave of the COVID-19 epidemic. The questionnaire is thorough and the response rate high given the circumstances. Although the results are solely descriptive, they do offer interesting perspective on ED preparations. There are some limitations worth addressing.

One major limitation of this study is that the reader is given little information as to who the respondents were and in what capacity they served during the COVID crisis. The authors state these individuals “may also have been the most involved professionals in crisis management in these EDs” (ln 206-207) without giving any evidence to the support this. Did a manager give the questionnaire to someone else to fill out? Were respondents aware of all the issues of care at the hospital? Self-report could affect the validity of responses, which needs to be addressed clearly as a limitation.

The authors then claim that this sample is representative for all Dutch ED (ln 169). How can the authors infer this? Are they able to compare characteristics of those who did versus did not response? Were there certain hospital characteristics associated with non-response (i.e. hospital density, timing of the first wave, burden of the epidemic)? This needs to be clearer.

The authors also conclude substantial heterogeneity in response (ln 43, ln 178-9), but which statistic is this based on? The increase in treatment capacity is likely dependent on several hospital level factors (hence the variation) and it seems that the hospitals had good reasons not to increase capacity. The rest of the statistics suggest fairly common strategies, with some departures due to sensible reasons. More specific discussion on these aspects is needed.

If the authors wanted to be more thorough, they could choose two or three important outcomes, gather hospital level covariates and responses to certain questions and perform a risk factor analysis. This could give more insight into the sources of the variation observed.

It was surprising to see that the date of finalization varied so much (ln 120). Could the authors give the distribution of these dates (i.e. median and 25 75th%tile)? And what were the reasons why some of the centers were delayed until May 2020 (which was practically the end of the first wave in the Netherlands)?

Finally, as the second/third wave of the Netherlands is coming to an end, how do these results relate to future epidemics? Any particular reflections?

Minor comments:

- Abstract. Information on the unit of analysis (i.e. one respondent per ED) and the total number of respondents needs to be reported.

- ln 35. “Alternative” should be defined here.

- ln 58. How are EDs “community-based resources”? Suggest a more appropriate term.

- ln 69. Is this supposed to be a separate paragraph?

- ln 72-73. Agree, but this claim is supported by what evidence?

- ln 75. Which body? RIVM? NVZ? NVSHA? Ref 6 only refers to the Amsterdam region.

- ln 139. Maintained for an unspecified amount of time?

- Table 2 & 3. Was n the same for all statistics presented?

- Table 2. Do the options under “No increase in treatment spaces” pertain to all respondents? Or those who did not increase treatment spaces? Needs to be clear.

- ln 162, ln 180. Not sure if this really is “crowding”. In the Dutch version, the authors explicitly asked if there was a shortage in personnel. Is this directly linked to crowding? Or because a large proportion of personnel became sick? The terms should be closer to the questions asked.

- ln 199-200. Is this based on personal feedback?

- ln 212. How does primary care help here? Were the majority of hospitalized patients referred to by the general practitioner in the Netherlands?

General comment:

- The authors should have the manuscript double-checked with a native speaker. There are some Dutchisms that crop up from time to time (e.g. the double subordinate clause on lns 72-73, “therewith” on ln 132, “plaster technician” in Table 3) or awkward phrasing (e.g. “hospitals’ crisis organizations” on ln 79, “contagious patient categories” on ln 141).

Reviewer #2: The authors describe a questionnaire-based study on changes in ED infrastructure, ED workforce adaptions and the role of emergency physicians in the hopsital’s crisis organization during the first wave of the COVID-19 pandemic in the Netherlands. All responding hospital organizations made preparations for a surge in COVID-19 patients to the ED.

Minor comments

* The authors got response from 66 of the 83 Dutch emergency departments. Percentages are presented in one decimal throughout the manuscript. Given the denominator (number of EDs), it would be more appropriate to only present integers without decimals.

* Abstract. Please consider to also mention the number of responding EDs and/or total number of Dutch EDs, next to the response rate, in the abstract.

* Methods-Study design, lines 91-92. Please consider changing ‘This was designed as’ into ‘This study was designed as’.

* Results-Changes in ED infrastructure, lines 139-142. Please add ‘in’ between ‘patient categories’ and ‘8 (13.8%)’.

* Discussion, lines 167-169. The authors state here that the results are representative because of the high response rate of 80%. I agree that it is an adequate response rate. Nevertheless, is it for example possible that no response was obtained from one or more regions or academic versus non-academic hospital organizations which could have affected the representativeness?

6. PLOS authors have the option to publish the peer review history of their article (what does this mean?). If published, this will include your full peer review and any attached files.

Reviewer #1: No

Reviewer #2: No

---

## [Author Response · Author response to Decision Letter 0]

4 Aug 2021

Response to reviewers’ comments:

Reviewer #1: In their manuscript, O'Connor et al describe the results of a questionnaire filled out by individuals involved in the management of Dutch emergency departments, specifically on the preparations involved during the first wave of the COVID-19 epidemic. The questionnaire is thorough and the response rate high given the circumstances. Although the results are solely descriptive, they do offer interesting perspective on ED preparations. There are some limitations worth addressing.

1. One major limitation of this study is that the reader is given little information as to who the respondents were and in what capacity they served during the COVID crisis. The authors state these individuals “may also have been the most involved professionals in crisis management in these EDs” (ln 206-207) without giving any evidence to the support this. Did a manager give the questionnaire to someone else to fill out? Were respondents aware of all the issues of care at the hospital? Self-report could affect the validity of responses, which needs to be addressed clearly as a limitation.

First, I want to thank both reviewers for their efforts to review the article. 

We agree that this point should be addressed. In the revised manuscript in the method section we have clarified that respondents were anonymous to the researchers. (lines 97-99) Furthermore, the discussion section is adapted according to the suggestions of the reviewer to highlight the limitation. (lines 230-231)

2. The authors then claim that this sample is representative for all Dutch ED (ln 169). How can the authors infer this? Are they able to compare characteristics of those who did versus did not response? Were there certain hospital characteristics associated with non-response (i.e. hospital density, timing of the first wave, burden of the epidemic)? This needs to be clearer.

We agree with the reviewer that the characteristics of EDs that did not respond would clarify to which extent the sample is representative for all Dutch EDs. We have therefore addressed the different characteristics the reviewer has suggested.

First, we added “The 16 EDs that did not respond were localized in different regions throughout the Netherlands and included university, teaching and peripheral hospitals.” (lines 125 – 126) and adjusted our claim in the discussion section to: ‘… the majority of …’ to the claim. (lines 183-184). 

Second, concerning the timing of the first wave, we have reason to believe that this was similar for both EDs that did and did not respond, since the EDs in both groups were spread throughout the country. We also added this information in the results section. (lines 125 – 126) 

Third, we added the regional incidence for all ED locations as of 31 May 2020, to table 1. (lines 129-133) 

This data is publicly available in the COVID-19 dataset from the Dutch National Institute for Public Health and the Environment. We have provided a histogram in the rebuttal letter to give insight in the regional incidence of COVID-19 of EDs that did and did not respond.

Upon analyzing the EDs that did not respond to the questionnaire, we discovered that one ED was closed as of January 2020, before COVID-19 reached the Netherlands. We therefore adjusted the total number of EDs from 83 to 82 in the introduction and results section. (line 87)

Histogram comparing participating EDs to EDs without a response regarding regional COVID-19 incidence

3. The authors also conclude substantial heterogeneity in response (ln 43, ln 178-9), but which statistic is this based on? The increase in treatment capacity is likely dependent on several hospital level factors (hence the variation) and it seems that the hospitals had good reasons not to increase capacity. The rest of the statistics suggest fairly common strategies, with some departures due to sensible reasons. More specific discussion on these aspects is needed.

We agree that the term ‘heterogeneity’ may have suggested the use of a statistic. We doubt whether a statistic exists that would be able to adequately measure heterogeneity in our population. We have therefore changed the section mentioned to: “This is reflected by the differences found between responding hospitals in this study”. (Line 194) 

It must also be noted that we discuss this further in the section following this sentence. As stated in our manuscript as well, we agree that hospitals may have had good reasons not to increase capacity. (lines 195 – 202)

4. If the authors wanted to be more thorough, they could choose two or three important outcomes, gather hospital level covariates and responses to certain questions and perform a risk factor analysis. This could give more insight into the sources of the variation observed. 

In our opinion, a risk factor analysis was not part of the primary research question. To be able to perform such an analysis, several extra parameters would have to be gathered. As our study was meant to be a descriptive study regarding the changes in ED infrastructure, ED workforce adaptations and the role of emergency physicians in the COVID crisis, we would like to refrain from performing such an additional analysis. We agree that offering additional explanations would indeed be interesting.

5. It was surprising to see that the date of finalization varied so much (ln 120). Could the authors give the distribution of these dates (i.e. median and 25 75th%tile)? And what were the reasons why some of the centers were delayed until May 2020 (which was practically the end of the first wave in the Netherlands)? 

We are thankful for this clever observation. After revising the data we discovered that the answer of 2 hospitals (finalization in May) probably had an erroneous answer in the questionnaire, because nearly all hospitals made changes to their ED infrastructure between the end of February and the end of March. The first wave was indeed practically the end of the first wave in the Netherlands. We have contacted these two EDs by telephone (emergency physician on duty) and they both provided dates in March (12 and 23 March, respectively). These dates were adjusted in the data set. We now also provide the distribution of the dates. The sentence was changed into: “The date when these preparations were completed varied between February 24 and April 1, 2020 (Median: 16 March 2020; IQR: 11 - 21 March 2020). (127 - 129)

6. Finally, as the second/third wave of the Netherlands is coming to an end, how do these results relate to future epidemics? Any particular reflections? 

We agree with the reviewer that the results of our study may also relate to future pandemics. We have added a paragraph with particular reflections in the discussion section: “As outbreaks of novel infectious diseases share similar characteristics, the results of this study may also relate to other future pandemics. Usually, little is known about the pathophysiology, symptomatology, and contagiousness of the disease. The lack of knowledge compels EDs to have a low threshold for isolating patients who might be contagious. EDs should therefore invest in isolation capacity. Furthermore, pandemics may result in high numbers of patients who require emergency care, which underlines the need for health care systems to have sufficient surge capacity. Finally, alterations of usual care, such as the redirection of low-acuity ED patients to outpatient departments, may help to alleviate the pressure on EDs during future pandemics.” (lines: 221 - 228)

Minor comments:

7. Abstract. Information on the unit of analysis (i.e. one respondent per ED) and the total number of respondents needs to be reported.

We added the information according to your suggestion. (lines 26 and 31)

8. ln 35. “Alternative” should be defined here.

“Such as outpatient departments” was added to the sentence. (lines 34-35)

9. ln 58. How are EDs “community-based resources”? Suggest a more appropriate term.

The term was replaced by “public health services”(line 61)

10. ln 69. Is this supposed to be a separate paragraph?

This was adjusted, as the reviewer correctly pointed out, the segment is part of the same paragraph. (lines 71 - 72)

11. ln 72-73. Agree, but this claim is supported by what evidence? (Gideon)

Considering the claim mentioned, we agree with the reviewer that it is poorly substantiated by the authors in the article we cited (Claim: “an individualized strategy for ED preparedness that considers baseline needs and available resources is superior to a blanket strategy for all EDs.”). We therefore added “Although this claim seems sensible, evidence is scant.” (lines 75-76)

12. ln 75. Which body? RIVM? NVZ? NVSHA? Ref 6 only refers to the Amsterdam region. 

This national body was the Landelijk Coördinatiecentrum Patiënten Spreiding. This was added to the text between brackets. (lines 78–79)

13. ln 139. Maintained for an unspecified amount of time? 

There was indeed no specified amount of time. The text has been adjusted. (lines 150-151)

14. Table 2 & 3. Was n the same for all statistics presented?

Percentages were recalculated and were adjusted when they did not correspond to the following: % = n (value) / n (total). An additional column was added to table 2 for “treatment places”. (lines 155-157 and 165-168)

15. Table 2. Do the options under “No increase in treatment spaces” pertain to all respondents? Or those who did not increase treatment spaces? Needs to be clear.

We adjusted table 2 according to the comments of the reviewer: “No increase in treatment spaces” was changed to “Number of treatment spaces was not increased during pandemic”. We also added “Reasons for not increasing treatment spaces:” and adjusted the indentation of the specific segment. Indentation of “Location of COVID-19 ED care”, “Location of non-COVID-19 ED care” and “Screening for COVID-19 before ED entry performed with” was also adjusted for clarity. (lines 155-157)

16. ln 162, ln 180. Not sure if this really is “crowding”. In the Dutch version, the authors explicitly asked if there was a shortage in personnel. Is this directly linked to crowding? Or because a large proportion of personnel became sick? The terms should be closer to the questions asked. (Gideon)

We agree with the reviewer that the English translation does not correspond to the question asked. We therefore adjusted the English Question 16 to:

“In the spring of 2020, during the peak of the pandemic in the Netherlands, was there a moment (or moments) when the capacity of the ED was insufficient?”. (Supplemental file 1, question 16) 

The comment regarding “crowding” made us realize we had not been clear which definition of ED crowding we referred to, as there have been multiple definitions in the past. To be clear to the reader which definition we use, we added “as described by Asplin et al.” to lines 176-177 and added the reference. This definition: “A situation in which the identified need for emergency services outstrips available resources in the ED.” is widely used in literature and corresponds to question 16.

17. ln 199-200. Is this based on personal feedback?

This was not based on personal feedback. As shown in Supplemental file 4, 9 respondents intended to maintain the improved interdisciplinary collaboration and 2 respondents the improved transmural collaboration. 

18. ln 212. How does primary care help here? Were the majority of hospitalized patients referred to by the general practitioner in the Netherlands? 

The majority of patients who present to Dutch EDs are referred by general practitioners or emergency medical services. We have added a sentence to make this clearer: “In the Netherlands, a relatively large proportion of ED patients (82%) is referred by a general practitioner or by emergency medical services.” (241-242)

General comment:

19. The authors should have the manuscript double-checked with a native speaker. There are some Dutchisms that crop up from time to time (e.g. the double subordinate clause on lns 72-73, “therewith” on ln 132, “plaster technician” in Table 3) or awkward phrasing (e.g. “hospitals’ crisis organizations” on ln 79, “contagious patient categories” on ln 141).

The whole manuscript was double-checked by a native speaker and alterations were made accordingly.

The Dutchisms mentioned by the reviewer were also adjusted: 

The double subordinate clause was addressed. (Lines 74-75)

“therewith shortening ED length of stay which” Was changed to: “The latter resulted in a shortened length of stay in the ED.” (144-145)

“hospitals’ crisis organizations” was altered to hospital’s crisis management team. (line 175)

“Plaster technician” was replaced by “orthopedic practitioner” (Table 2 lines 155-157). This was also corrected in the English version of the questionnaire (supplemental file 1) and the dataset.

Reviewer #2: The authors describe a questionnaire-based study on changes in ED infrastructure, ED workforce adaptions and the role of emergency physicians in the hopsital’s crisis organization during the first wave of the COVID-19 pandemic in the Netherlands. All responding hospital organizations made preparations for a surge in COVID-19 patients to the ED.

Minor comments

20. The authors got response from 66 of the 83 Dutch emergency departments. Percentages are presented in one decimal throughout the manuscript. Given the denominator (number of EDs), it would be more appropriate to only present integers without decimals.

We agree with the reviewer and adjusted the percentages regarding number of EDs throughout the manuscript so only integers without decimals are presented.

21. Abstract. Please consider to also mention the number of responding EDs and/or total number of Dutch EDs, next to the response rate, in the abstract.

We added ”The questionnaire was completed by 58 respondents, covering” to the abstract so the number of responding EDs is mentioned next to the response rate. (line 31)

22. Methods-Study design, lines 91-92. Please consider changing ‘This was designed as’ into ‘This study was designed as’.

We adjusted the sentence according to the comment. (line 94)

23. Results-Changes in ED infrastructure, lines 139-142. Please add ‘in’ between ‘patient categories’ and ‘8 (13.8%)’.

We adjusted the sentence according to the comment. (line 153)

24. Discussion, lines 167-169. The authors state here that the results are representative because of the high response rate of 80%. I agree that it is an adequate response rate. Nevertheless, is it for example possible that no response was obtained from one or more regions or academic versus non-academic hospital organizations which could have affected the representativeness?

We agree with the reviewer that this claim deserves attention. As we responded to reviewer 1, we added characteristics of EDs that did not respond to the results (lines 125 -126) table 1 in the results (lines 129-133). We also adjusted our claim accordingly (182 – 184).

---

## [Decision Letter · Decision Letter 1]

10 Aug 2021

PONE-D-21-12869R1

Preparations of Dutch emergency departments for the COVID-19 pandemic: a questionnaire-based study.

PLOS ONE

Dear Dr. O'Connor,

Thank you for submitting your manuscript to PLOS ONE. After careful consideration, we feel that it has merit but does not fully meet PLOS ONE’s publication criteria as it currently stands. Therefore, we invite you to submit a revised version of the manuscript that addresses the points raised during the review process.

Please revise accordingly.

We look forward to receiving your revised manuscript.

Kind regards,

Academic Editor

PLOS ONE

Journal Requirements:

Reviewers' comments:

Reviewer's Responses to Questions

**Comments to the Author**

1. If the authors have adequately addressed your comments raised in a previous round of review and you feel that this manuscript is now acceptable for publication, you may indicate that here to bypass the “Comments to the Author” section, enter your conflict of interest statement in the “Confidential to Editor” section, and submit your "Accept" recommendation.

Reviewer #1: All comments have been addressed

Reviewer #2: All comments have been addressed

2. Is the manuscript technically sound, and do the data support the conclusions?

Reviewer #1: (No Response)

Reviewer #2: Yes

3. Has the statistical analysis been performed appropriately and rigorously? 

Reviewer #1: (No Response)

Reviewer #2: Yes

4. Have the authors made all data underlying the findings in their manuscript fully available?

Reviewer #1: (No Response)

Reviewer #2: Yes

5. Is the manuscript presented in an intelligible fashion and written in standard English?

Reviewer #1: (No Response)

Reviewer #2: Yes

6. Review Comments to the Author

Reviewer #1: The responses are clear and I thank the authors for incorporating many of my suggestions.

Two very minor issues:

- "Plaster technicians" is still in Table 3.

- ln 178. The definition of crowding should be mentioned in the methods, not here.

Reviewer #2: (No Response)

7. PLOS authors have the option to publish the peer review history of their article (what does this mean?). If published, this will include your full peer review and any attached files.

Reviewer #1: No

Reviewer #2: No

---

## [Author Response · Author response to Decision Letter 1]

17 Aug 2021

1. Reference list: 

The reference list was reviewed and minor adjustments to the style were made to meet PLOS requirements. None of the references were removed or retracted.

Response to reviewers’ comments:

We would like to thank the reviewers for their thoughtful comments and efforts towards improving our manuscript. The minor issues were adjusted according to their suggestions.

Reviewer #1: The responses are clear and I thank the authors for incorporating many of my suggestions. Two very minor issues:

• "Plaster technicians" is still in Table 3.

‘Plaster technician’ was alterered to ’Orthopedic practitioner’

• ln 178. The definition of crowding should be mentioned in the methods, not here.

The sentence was moved to lines 107-108 of the methods.

---

## [Decision Letter · Decision Letter 2]

20 Aug 2021

Preparations of Dutch emergency departments for the COVID-19 pandemic: a questionnaire-based study.

PONE-D-21-12869R2

Dear Dr. O'Connor,

We’re pleased to inform you that your manuscript has been judged scientifically suitable for publication and will be formally accepted for publication once it meets all outstanding technical requirements.

Kind regards,

Academic Editor

PLOS ONE

Additional Editor Comments (optional):

Reviewers' comments:

Reviewer's Responses to Questions

**Comments to the Author**

1. If the authors have adequately addressed your comments raised in a previous round of review and you feel that this manuscript is now acceptable for publication, you may indicate that here to bypass the “Comments to the Author” section, enter your conflict of interest statement in the “Confidential to Editor” section, and submit your "Accept" recommendation.

Reviewer #1: (No Response)

Reviewer #2: All comments have been addressed

2. Is the manuscript technically sound, and do the data support the conclusions?

Reviewer #1: (No Response)

Reviewer #2: (No Response)

3. Has the statistical analysis been performed appropriately and rigorously? 

Reviewer #1: (No Response)

Reviewer #2: (No Response)

4. Have the authors made all data underlying the findings in their manuscript fully available?

Reviewer #1: (No Response)

Reviewer #2: (No Response)

5. Is the manuscript presented in an intelligible fashion and written in standard English?

Reviewer #1: (No Response)

Reviewer #2: (No Response)

6. Review Comments to the Author

Reviewer #1: (No Response)

Reviewer #2: (No Response)

7. PLOS authors have the option to publish the peer review history of their article (what does this mean?). If published, this will include your full peer review and any attached files.

Reviewer #1: No

Reviewer #2: No

---

## [Editor Report · Acceptance letter]

31 Aug 2021

PONE-D-21-12869R2 

Preparations of Dutch emergency departments for the COVID-19 pandemic: a questionnaire-based study. 

Dear Dr. O'Connor:

I'm pleased to inform you that your manuscript has been deemed suitable for publication in PLOS ONE. Congratulations! Your manuscript is now with our production department. 

Kind regards, 

on behalf of

Dr. Robert Jeenchen Chen 

Academic Editor

PLOS ONE